# Multiplexed imaging of nucleome architectures in single cells of mammalian tissue

Miao Liu[1,4], Yanfang Lu[1,4], Bing Yang [1], Yanbo Chen[1], Jonathan S. D. Radda [1], Mengwei Hu [1], Samuel G. Katz[2] & Siyuan Wang [1,3✉]

The three-dimensional architecture of the genome affects genomic functions. Multiple genome architectures at different length scales, including chromatin loops, domains, compartments, and lamina- and nucleolus-associated regions, have been discovered. However, how these structures are arranged in the same cell and how they are mutually correlated in different cell types in mammalian tissue are largely unknown. Here, we develop Multiplexed Imaging of Nucleome Architectures that measures multiscale chromatin folding, copy numbers of numerous RNA species, and associations of numerous genomic regions with nuclear lamina, nucleoli and surface of chromosomes in the same, single cells. We apply this method in mouse fetal liver, and identify de novo cell-type-specific chromatin architectures associated with gene expression, as well as cell-type-independent principles of chromatin organization. Polymer simulation shows that both intra-chromosomal self-associating interactions and extra-chromosomal interactions are necessary to establish the observed organization. Our results illustrate a multi-faceted picture and physical principles of chromatin organization.

[1] Department of Genetics, Yale School of Medicine, Yale University, New Haven, CT 06510, USA. [2] Department of Pathology, Yale School of Medicine, Yale University, New Haven, CT 06510, USA. [3] Department of Cell Biology, Yale School of Medicine, Yale University, New Haven, CT 06510, USA. [4] These authors contributed equally: Miao Liu, Yanfang Lu. ✉email: siyuan.wang@yale.edu

I n the mammalian cell nucleus, DNA folds into functional spatial architectures across multiple length scales[1–7]. The first level of DNA folding involves the wrapping of genomic DNA around histone proteins to form individual nucleosomes—the basic structural unit of chromatin. Interspaced regions of the chromatin may interact, forming loop structures such as promoter-enhancer loops that are involved in gene activation[1–4,7]. At the other end of the spectrum, individual chromosomes occupy distinct nuclear space, known as chromosomal territories[6]. Recently, sequencing-based chromosome-conformation-capture methods, such as Hi–C[4,8], have revealed two types of intermediate structures known as topologically associating domains (TADs, also known as contact domains) and A/B compartments[8–13]. TADs are consecutive self-interacting genomic regions each containing tens to hundreds of kilobases (kb) of DNA[9–12]. Compartments A and B each contain multiple TADs and are enriched with active (A) and inactive (B) chromatin, respectively[8,13]. Other sequencing methods have also been used to identify genomic regions adjacent/attached to the nuclear lamina or nucleolus, known as lamina-associated domains (LADs) and nucleolus-associated chromatin domains (NADs), both of which are associated with transcriptional inactivation[14–16].

It remains largely unclear, however, how these diverse nucleome architectures are jointly organized in the same single cells, and how they correlate with each other across heterogeneous cell populations in mammalian tissues. Bulk chromosome-conformation-capture methods cannot distinguish cell-type-specific genome architectures in a mixed population. While recent advances in single-cell Hi–C and related methods have significantly improved the genomic resolution of the technique at the single-cell level[17], they have not enabled single-cell mapping in tissue. Furthermore, single-cell Hi–C has not been combined with profiling of RNA expression, LADs, and NADs in the same single cells. In contrast, fluorescence microscopy approaches offer direct, single-cell visualization of many cellular structures[18–20]. Conventional DNA fluorescence in situ hybridization (FISH), for example, allows direct mapping of the spatial positions of two or more genomic loci in single cells (e.g. ref. [21]). Recent sequential FISH techniques increased the number of genomic loci mapped in single cells and allowed direct tracing of chromatin folding in mammalian cell cultures and *Drosophila* embryos[22–26]. However, multiscale chromatin tracing from promoter-enhancer loops to whole chromosomes, with simultaneous profiling of transcripts, lamina, and nucleolar associations, has not been achieved. Furthermore, chromatin tracing in mammalian tissue has not been accomplished.

To address these limitations and enable analysis of multiscale nucleome architectures in heterogeneous mammalian tissue in a cell-type-specific manner, here we develop Multiplexed Imaging of Nucleome Architectures (MINA)—an integrative method capable of single-cell, in situ measurements of multiscale chromatin folding across four orders of magnitude of genomic length, proximity of numerous genomic loci to lamina and nucleoli, and RNA copy numbers from over one hundred genes (Fig. 1a). We apply this technique to study single-cell nucleome architectures and gene expression in the distinct cell types of E14.5 mouse fetal liver (Fig. 1a). First, to test the capability of this method to resolve cell-type specific chromatin folding, we study the 3D folding of chromatin at the promoter-enhancer and TAD-to-chromosome length scales in single cells in fetal liver, and distinguish different cell types based on their RNA profiles. We demonstrate de novo discovery of cell-type-specific chromatin folding schemes at these length scales, and show that chromatin folding differences at both scales are correlated with gene expression changes between cell types. Next, to demonstrate the ability of this method to probe the joint organization and co-variation of multiple nucleome architectures, we examine the correlations between chromatin folding

and the association of chromatin with nuclear lamina, nucleoli, and the surface of the chromosome territory in the different cell types. We observe both cell-type-specific features and cell-type-invariant principles of the joint organization of nucleome architectures. Finally, we build a polymer model to computationally simulate and explain the observed correlations between nucleome architectural features. We find that intra-chromosomal self-associating interactions are insufficient to explain the observed chromosome architectures, and that both intra-chromosomal and extra-chromosomal interactions are required to establish the observed features.

## Results

**Development of the MINA method.** MINA involves multiscale chromatin tracing, measurements of lamina and nucleolar associations, and highly multiplexed RNA imaging. To trace multiscale chromatin folding, we hybridized a library of primary oligonucleotide probes to mouse fetal liver tissue sections, labeling the central 100-kb regions of 50 TADs along the entire mouse chromosome 19 (Chr19), as well as 19 consecutive 5-kb segments upstream of the gene *stearyl-CoA desaturase 2* (*Scd2*) located on Chr19 (Fig. 1a). *Scd2* is known to be expressed in hepatocytes in fetal liver, and is critical for lipid synthesis during early liver development[27]. The 19 segments span multiple potential cis-regulatory elements[28] with unknown folding structure. Each oligonucleotide probe in the primary probe library contained a unique genomic sequence that hybridizes to the targeted genomic region, and a nongenomic readout sequence shared by all primary probes targeting the same genomic segment (Fig. 1b). The readout region can hybridize to dye-labeled secondary probes with complementary sequences in a series of secondary hybridizations to sequentially visualize the 3D positions of the labeled genomic regions (Fig. 1b–d). The 3D folding of chromatin can be reconstructed by linking these positions into traces (Fig. 1c, d). We used 69 secondary probes to distinguish all 69 probed genomic regions. This approach is conceptually similar to our previous chromatin tracing method[26] and subsequent versions from others[22–25], but has significantly expanded the range of genomic length scales probed in the same experiment to over four orders of magnitude (from 5 kb to over 50 Mb), with improved probe design and FISH procedure to allow tracing at 5-kb resolution and in mouse tissue in combination with other modes of imaging. To measure the proximity of genomic loci to the nucleoli and nuclear lamina, we labeled nucleoli by immunofluorescence staining of fibrillarin and imaged whole nuclei with SYTOX or DAPI stain (Fig. 1e). We approximated nuclear lamina locations as the boundaries of the nuclei. To efficiently image and distinguish over 100 RNA species with single-molecule resolution, we adapted RNA multiplexed error-robust FISH (MERFISH)[29,30] to fetal liver tissue (Fig. 1f). We probed 137 RNA species in total, 55 of which were expressed from marker genes of major cell types in fetal liver[31]. The other 82 were from genes located on Chr19. Each RNA species was labeled with primary oligonucleotide probes containing targeting sequences complementary to different parts of the RNA, and a unique combination of 4 out of 16 readout sequences. This combination formed a unique barcode for the RNA species (Fig. 1f). We imaged and read out this barcode with single molecule resolution by sequentially applying 16 dye-labeled secondary probes that hybridized to the readout sequences (Fig. 1f, g).

To segment the tightly packed fetal liver cells and allow quantification of RNA copy numbers in single cells, we labeled cell boundaries with oligo-conjugated wheat germ agglutinin (WGA), and visualized the WGA pattern with an additional dye-labeled secondary probe hybridized to the WGA-conjugated oligo

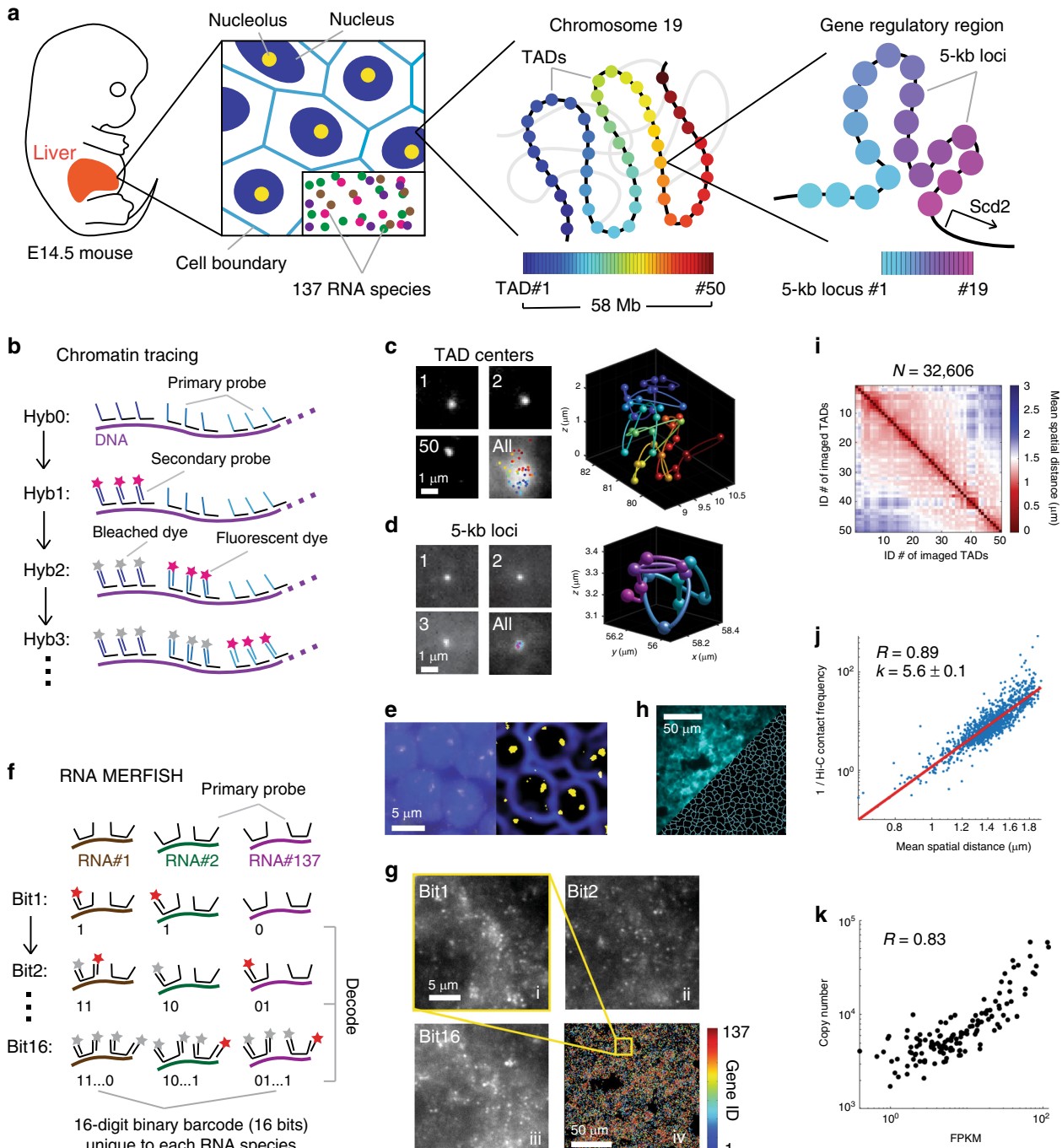

**Fig. 1 Mapping nucleome architectures in single cells of mammalian tissue. a** Schematic illustration of the biological features measured by Multiplexed Imaging of Nucleome Architectures (MINA). We imaged cell boundaries, nuclei, nucleoli, 137 different RNA species, 50 TADs in chromosome 19 (Chr19), and 19 consecutive 5-kb loci upstream of gene *Scd2* in E14.5 mouse fetal liver tissue sections. **b** A simplified scheme of the chromatin tracing approach. All genomic regions were first labeled with primary probes (Hyb0), and then sequentially visualized with dye-labeled secondary probes (Hyb1, 2, 3…). **c, d** (Left panels) Individual and sum images of targeted TADs (**c**) or loci (**d**). Images are max projections along the z direction of the 3D image stacks. (Right panels) 3D positions of targeted regions plotted as pseudo-colored spheres connected with a smooth curve. **e** Raw (left panel) and processed (right panel) images of cell nuclei (blue) and nucleoli (yellow). **f** A simplified scheme of the RNA profiling approach. Primary probes were first hybridized to the RNA molecules, which encoded each RNA species with a unique 16-bit barcode. Then the barcode was decoded by sequentially visualizing the bits. **g** (i-iii) Images of RNA molecules in three rounds of secondary hybridization. Images are from a single z position in the 3D image stacks. (iv) All identified RNA molecules in a field of view pseudo-colored based on their gene identities. The yellow boxed region is the same region shown in i-iii. **h** Raw (top left) and processed (bottom right) images of cell boundaries. **i** Mean spatial distance matrix of the 50 TADs, with each element showing the mean spatial distance between a pair of TADs. **j** Inverse Hi–C contact frequency versus mean spatial distance for each pair of TADs. Each dot represents a pair of TADs. **k** Total RNA copy numbers from imaging versus FPKM values from bulk RNA sequencing for each probed RNA species. Data from 137 RNA species were used to generate (**k**). Results in Fig. 1 are representative of four biological replicates. Source data are provided as a Source data file.

(Fig. 1h). The centroid positions of imaged genomic regions, the positions of nucleoli, the boundaries of cells and nuclei, and the barcodes of each RNA molecules were computationally extracted from the 3D image stacks (Fig. 1c-h). We routinely imaged and analyzed thousands of cells per tissue section during the two consecutive days of imaging involved in each full MINA experiment.

**Validation of MINA measurements**. To validate MINA measurements, we compared our imaging results with available sequencing data. First, we calculated the mean spatial distance between each pair of imaged TADs, and obtained 1225 pair-wise distances (Fig. 1i). We compared these distances with the corresponding contact frequencies between the TADs measured by ensemble-average Hi–C, which offered enough resolution for measuring chromatin interactions at the TAD-to-chromosome scale in E14.5 mouse fetal liver[32] (Supplementary Fig. 1A). The mean spatial distances were highly correlated with the inverse contact frequencies, with a Pearson correlation coefficient of 0.89 (Fig. 1j). Our analysis showed the Hi–C contact frequency was inversely proportional to the 5th power of the mean spatial distance (Fig. 1j). This power-law relationship between Hi–C contact frequency and mean spatial distance is similar to that previously obtained from human cell cultures[22,26]. We then compared our RNA MERFISH results with bulk RNA sequencing data from E14.5 mouse fetal liver. The RNA copy numbers of the 137 probed genes counted from all imaged cells showed a high correlation with the RNA abundance measured by sequencing, with a Pearson correlation coefficient of 0.83 (Fig. 1k). These high correlations between our imaging measurements and those from entirely different methods provided a validation of our technique.

Next, we used a normalization and compartmentalization analysis procedure previously introduced to determine the A/B compartments from the mean spatial distance matrix[26]. This analysis yielded population-averaged compartment scores of TADs, which reflected their extent of association with compartment A (positive scores) or B (negative scores) (Supplementary Fig. 1B–E). Concurrently, we quantified the probabilities of different TADs being in spatial proximity to nucleoli or nuclear lamina, and termed these probabilities the nucleolar association ratios and lamina association ratios, respectively (TADs within 200 nm of the abstracted nucleolar or perinuclear voxels were considered as being associated with nucleoli or nuclear lamina). Our analyses showed that A/B compartment scores were negatively correlated with lamina association ratios and nucleolar association ratios (Supplementary Fig. 1F–G), consistent with the enrichment of inactive chromatin in LADs and NADs previously shown in cell cultures[14–16].

**De novo chromatin folding patterns in mammalian tissue**. To identify different cell types and, in turn, cell-type-specific chromatin architectures, we clustered individual cells based on the similarities of their single-cell RNA copy number profiles (Fig. 2a). Based on the enrichment of different cell-type marker transcripts in the clusters, we identified seven major cell types in fetal liver, including hepatocytes, erythroblasts, proerythroblasts, macrophages, endothelial cells, megakaryocytes and a combined cluster of other cell types (Fig. 2a, b). The expression of *Scd2* was specifically enriched in hepatocytes (Fig. 2c). Several enhancers have been annotated upstream of *Scd2*[28], but which one(s) of them interact(s) with the promoter of *Scd2* in fetal liver hepatocytes, or whether any one of them interacts with the promoter at all, is unknown. We characterized the fine folding structure of chromatin upstream of the *Scd2* gene in hepatocytes versus all other cell types by plotting mean spatial distance matrices of the

5-kb-resolution chromatin traces (Fig. 2d). The matrices revealed a decreased distance from traced region 16 to the promoter region 19 in hepatocytes (Fig. 2d). Region 16 contains one of the annotated enhancers (Fig. 2e). We then defined genomic regions with a spatial distance below 150 nm as being in contact with each other, and measured the contact probability of each traced region with region 19. The contact probability between regions 16 and 19 increased in hepatocytes in comparison to non-hepatocytes (Fig. 2f). These results suggest an interaction between the enhancer in region 16 and the *Scd2* promoter in hepatocytes.

Next, we asked whether the A/B compartmentalization schemes differed among the fetal liver cell types, and whether any compartmentalization differences could explain the changes in expression of the probed Chr19 genes. To answer these questions, we grouped large-scale chromatin traces of the 50 TADs by cell types, and determined the A/B compartmentalization of TADs in each cell type (Fig. 3a, b, Supplementary Fig. 2). The compartment scores of TADs varied between cell types (Fig. 3b, Supplementary Fig. 2), and the changes in compartment scores were associated with changes in gene expression. When comparing one cell type to another, a significant (more than three-fold) increase in the expression level of a Chr19 gene was more likely to be coupled with an increase in the compartment score of the TAD containing the gene than with a decrease in the compartment score (Fig. 3c). However, significant differences in compartment scores did not guarantee significant changes in gene expression (Fig. 3c). These observations suggest that A/B compartmentalization may serve as an additional layer of control over gene expression and cell identity, but also that it is not the sole determinant of expression differences between cell types in developing mouse liver.

Our previous study on a human cell line revealed that A/B compartments are organized in a polarized, side-by-side manner in individual chromosomes[26] (Fig. 3d top panel). It is unclear whether this is a cell-line-specific or human-specific feature, or whether this principle is also conserved in mouse Chr19 in the different cell types in fetal liver tissue. To address this question, we visualized the spatial arrangement of A/B compartments in individual chromosomes (Fig. 3d bottom panel), and measured the polarization index of the A/B compartments, previously defined as the geometric mean of the non-overlapping proportions of two 3D convex hulls surrounding the two compartments[26] (Supplementary Fig. 3A). Under this definition, if compartments A and B are completely separated and are positioned in a polarized fashion, the polarization index should equal one. If one compartment wraps around the other, or if they completely overlap, the polarization index should equal zero. Our data showed that the polarization indices from individual Chr19s in different cell types are always significantly higher than the control values with randomized compartment assignments (Fig. 3e and Supplementary Fig. 3B). Consistently, when individual chromosomes were aligned along a vector that points from the center of compartment A to the center of compartment B, the distribution of compartment A TADs showed a significant displacement relative to the B center, and vice versa (Supplementary Fig. 4). This suggests that the polarized organization of compartments A and B is conserved in mouse Chr19 in most fetal liver cells, regardless of cell type.

**Principles of joint nucleome architectures**. The integrative nature of MINA allowed us to investigate the intricate relationship among multiple features of nucleome organization in a cell-type-specific manner in tissue. A recent report on mouse embryonic fibroblasts showed that LADs largely correspond to compartment B domains[33]. Here we asked whether the

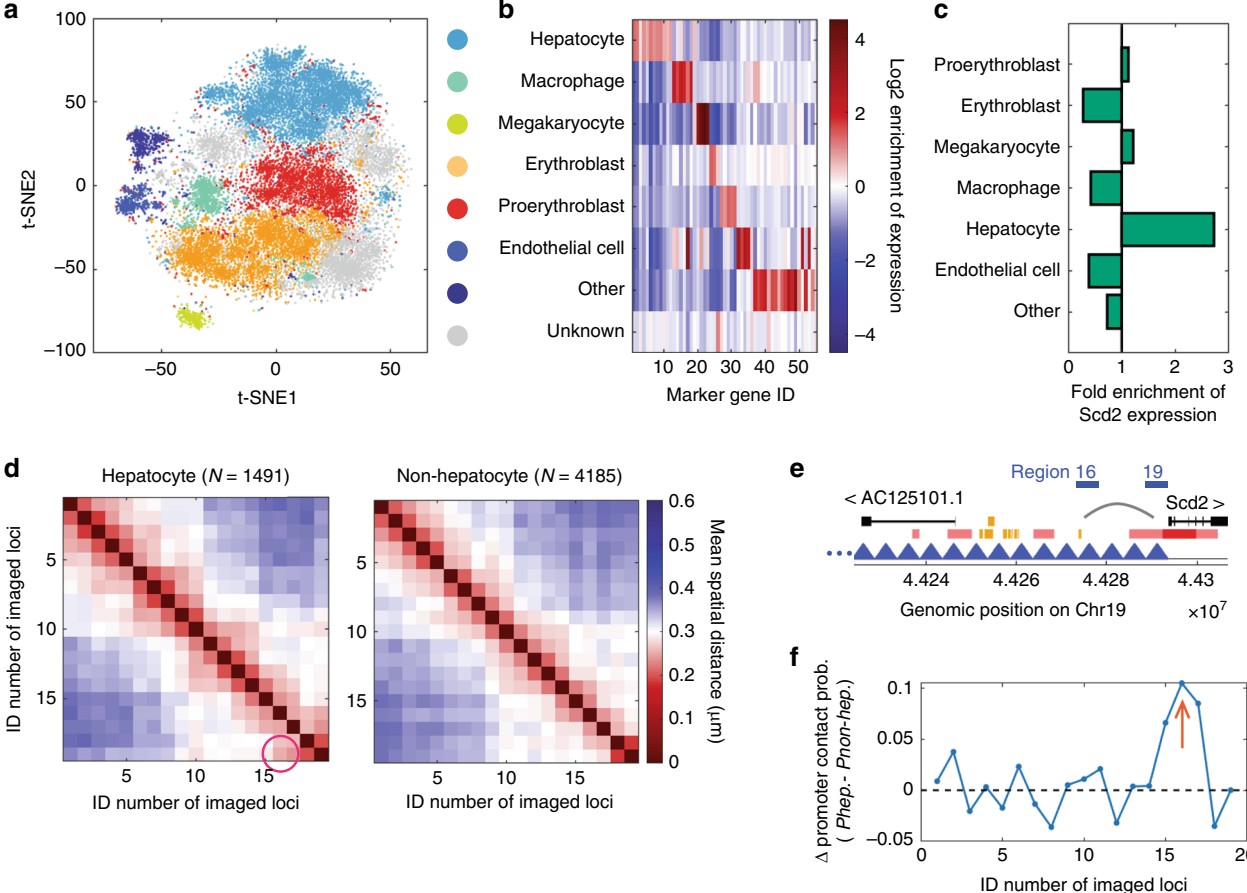

**Fig. 2 Cell-type-specific chromatin folding at the promoter-enhancer length scale. a** t-distributed stochastic neighbor embedding (t-SNE) plot of single-cell RNA profiles with pseudo-colored clusters that correspond to different cell types. **b** Average gene expression profiles of probed cell-type marker genes for the different clusters in A. Cell types were identified based on the enrichment of different marker genes. **c** Fold enrichment of *Scd2* RNA copy numbers in different cell types. **d** Mean spatial distance matrices of the 19 5-kb loci upstream of *Scd2* in hepatocytes (left) and non-hepatocytes (right). The magenta circle highlights a decreased distance in hepatocytes that suggests looping contact between the corresponding loci. **e** Gene regulatory annotations of the genomic region upstream of *Scd2*. Red: promoters. Pink: promoter flank regions. Orange: enhancers. Blue: probed 5-kb regions. **f** Difference in contact probabilities of each locus with *Scd2* promoter (locus 19) in hepatocytes versus non-hepatocytes. Arrow: contact between locus 16 and *Scd2* promoter. Source data are provided as a Source data file.

compartmentalization differences among the fetal liver cell types could largely explain their differences in the lamina association ratios and nucleolar association ratios of TADs. Our data showed that the lamina/nucleolar association ratios varied among the fetal liver cell types even when the compartmentalization differences were taken into account (Fig. 4a, b). For example, both the nucleolar and lamina association ratios in proerythroblasts were systematically higher than those in the closely related erythro-blasts across the full range of compartment scores (Fig. 4a, b). In addition, the extent of the correlations between lamina/nucleolar association ratios and the compartment scores also varied between cell types (Fig. 4c, Supplementary Figs. 5 and 6). For example, erythroblasts showed a stronger correlation between compartment scores and lamina association ratios than did proerythroblasts (Fig. 4c), even though the latter had higher values of lamina association ratios. These observations suggest that compartment scores alone are not sufficient to explain the extent of association with nuclear lamina and nucleoli in different cell types.

We next asked whether the polarized organization of A/B compartments in individual chromosomes depends on the association of compartment B with nuclear lamina or nucleoli. To address this question, we grouped individual chromosomes

based on whether their compartment B is associated with nuclear lamina and nucleoli, which yielded four groups: chromosomes associated with neither lamina nor nucleoli, chromosomes associated with lamina but not nucleoli, chromosomes associated with nucleoli but not lamina, and chromosomes associated with both lamina and nucleoli. We found similar polarization indices in all four groups (Fig. 4d, Supplementary Fig. 7). This indicated that the polarized organization of A/B compartments did not depend on lamina or nucleolar association.

Furthermore, we analyzed how A/B compartment scores might be correlated with the probabilities of TADs being localized to the surface of the chromosome territory, termed chromosome surface ratios. Multiple recent reports suggested phase-separation (or in general, self-associations between chromatin regions with the same epigenetic content or transcription machinery) as an important mechanism to drive the interactions of active/inactive chromatin[34–38]. Based on a pure phase-separation or self-association model, one might expect TADs with stronger compartment identities to form a hub of chromatin interactions, and thus may localize to the interior of the chromosome territory. To determine whether each TAD was located at the surface or interior of its chromosome territory, we built a 3D convex hull of the chromosome based on the positions of the imaged TADs. If a

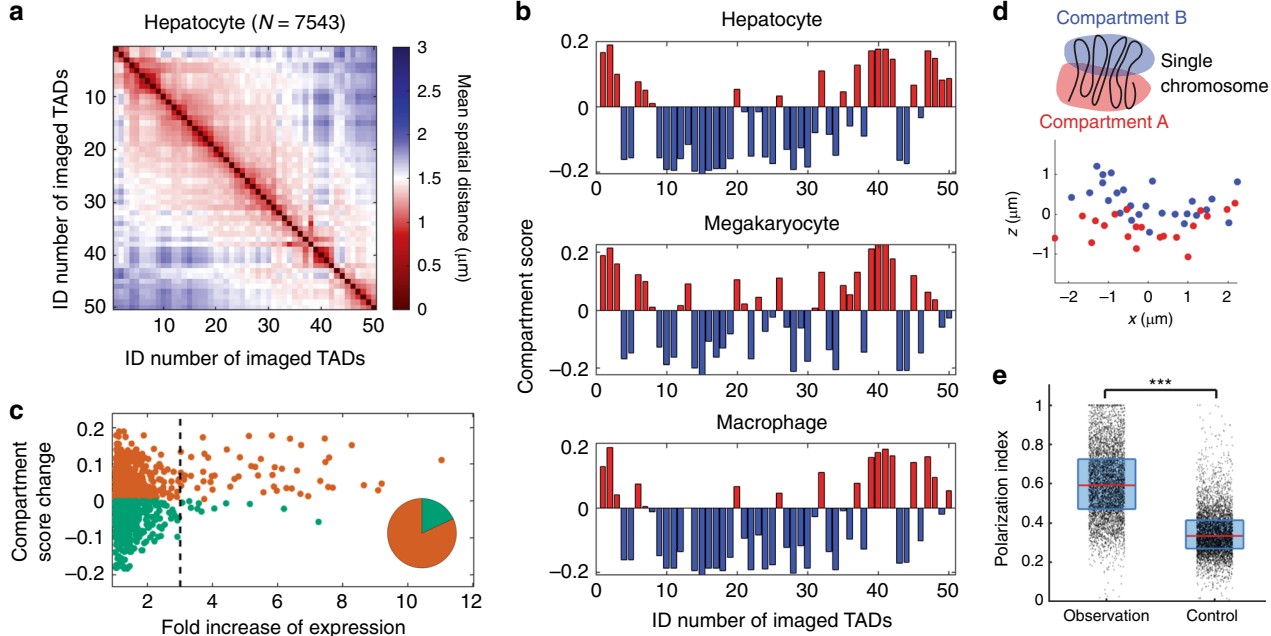

**Fig. 3 Cell-type-specific chromatin folding at the TAD-to-chromosome length scale. a** Mean spatial distance matrix of the 50 TADs in hepatocytes. **b** Compartment scores of the 50 TADs in three cell types. **c** Fold increase of RNA copy number for each gene between each pair of cell types versus the change of compartment score of the gene region between the pair of cell types. Pie chart: Counts of data points with more than three-fold increase of expression. Orange: increase of compartment score. Green: decrease of compartment score. **d** (Top panel) Schematic illustration of the concept of compartment polarization. (Bottom panel) Spatial positions of compartment-B TADs (blue dots) and compartment-A TADs (red dots) in a copy of Chr19. The chromosome was rotated in space for better visualization of the polarized organization of A/B compartments. **e** Polarization indices of individual chromosomes in proerythroblasts. Observed values were compared with a randomization control, where we randomized the compartment assignments of TADs while maintaining the number of TADs in each compartment. Each dot corresponds to a single copy of Chr19. Data from 4830 chromosomes were used to generate each observation and control group in **e**. The red lines represent medians. The boxes show the 25% and 75% quantiles. ***$p < 10^{-307}$ (two-sided Wilcoxon rank sum test; the exact p value is smaller than the smallest positive double precision floating-point number in MATLAB). Observation minimum, 25% quantile, median, 75% quantile, and maximum equal 0.02, 0.47, 0.59, 0.73, and 1, respectively. Control minimum, 25% quantile, median, 75% quantile, and maximum equal to 0.01, 0.27, 0.33, 0.41, and 1 respectively. Source data are provided as a Source data file.

TAD was on the surface of the 3D convex hull, we considered that TAD as being localized to the chromosome surface. Our data showed that TADs in compartments A and B had similar ranges of chromosome surface ratios (Fig. 5), consistent with the polarized organization of A/B compartments. However, the compartment A scores alone were positively correlated with the chromosome surface ratios, whereas the compartment B scores were negatively correlated with the chromosome surface ratios, although the strength of the latter correlation was often moderate (Fig. 5). This observation suggests that contrary to the expectation above, TADs with stronger compartment identities tend to localize to the surface of the chromosome territory, while TADs with weaker or more ambiguous compartment identities tend to localize to the interior of the chromosome territory.

**Modeling of chromatin interactions.** Our experiments showed that A and B compartments of mouse Chr19 are organized in a polarized manner, and that TADs with stronger compartment strength have the tendency of chromosome surface localization. To explore the physical mechanism underlying our observed chromosome organization, we built a minimal polymer model of Chr19 and investigated what chromatin interactions are necessary to establish the following features: a polarized organization of A and B compartments, a positive correlation between compartment A scores and chromosome surface ratios, and a moderately negative correlation between compartment B scores and chromosome surface ratios. We computationally simulated the spatial movement of a polymer with 50 monomers representing the 50 TADs, and modeled A/B compartmentalization as a self-

association process (Fig. 6). We assumed that compartment scores quantitatively reflect the enrichment of self-associating factors, and used the measured compartment scores from hepatocyte Chr19 for the simulation. A recent report suggested that chromatin interactions in the B compartment are crucial for the spatial separation of A/B compartments[39]. In our simulation, we started by having self-associating interactions only among compartment-B TADs in single chromosomes (B–B interaction). In this simulation, each TAD in compartment B favors other compartment-B TADs in its vicinity, with an energy decrease proportional to the compartment B scores of the interacting TADs. The simulation showed that under this setting, compartment A largely wrapped around compartment B, forming a radial organization (Fig. 6a). The polarization index of this organization was significantly lower than the randomization control (Fig. 6b), and the chromosome surface ratios were significantly correlated with the overall A/B compartment scores, with compartment-A TADs having higher chromosome surface ratios (Fig. 6c). These results were contrary to our experimental observations and suggest that intra-chromosomal B–B interaction alone is insufficient to explain the polarized organization of A/B compartments. Next, we added A–A interactions to the model in a similar fashion as B–B interactions. This addition led to a polarized organization of the A/B compartments in the simulated traces, and a lack of an overall correlation between A/B compartment scores and chromosome surface ratios (Fig. 6d–f), both consistent with our experimental observations. However, when TADs in compartments A and B were analyzed separately, compartment A scores were negatively correlated with chromosome surface ratios,

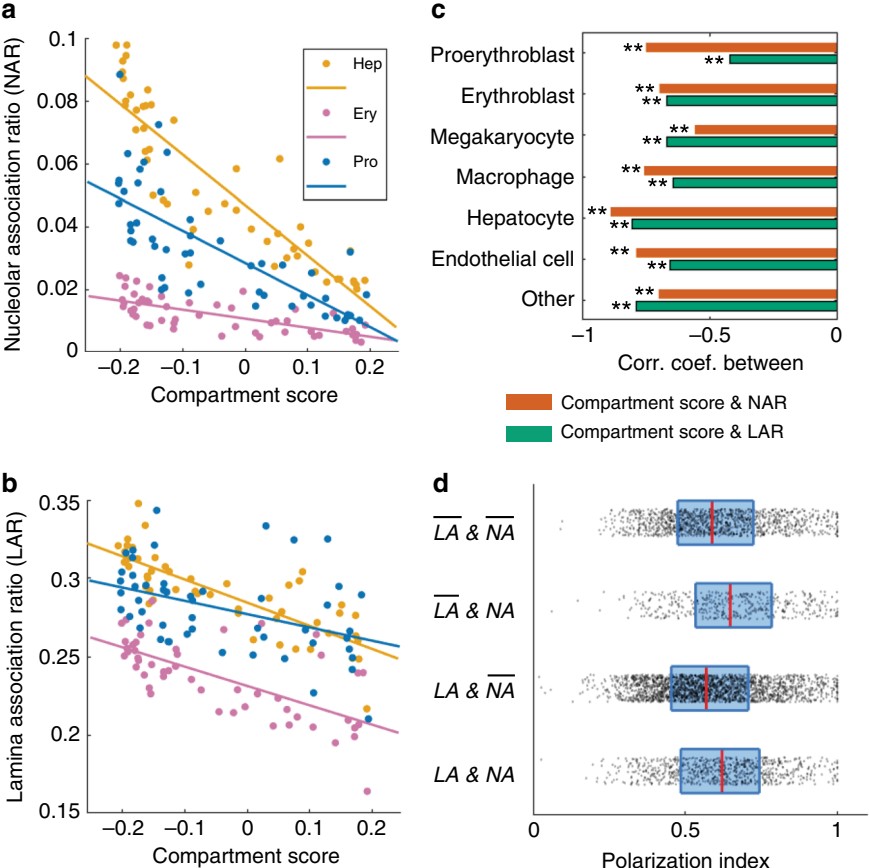

**Fig. 4 Correlations between multiple nucleome architectural features. a** Nucleolar association ratios of TADs versus their compartment scores in hepatocytes (Hep), erythroblasts (Ery) and proerythroblasts (Pro). **b** Lamina association ratios of TADs versus their compartment scores in the three cell types above. **c** Correlation coefficients between compartment scores and lamina or nucleolar association ratios in all identified cell types. **p = (top to bottom) 4e-10, 2e-3, 2e-8, 1e-7, 2e-5, 1e-7, 2e-10, 4e-7, 7e-18, 2e-12, 1e-11, 2e-7, 1e-8, 1e-11. The p values were calculated for Pearson's correlation using a two-sided Student's t distribution. No adjustment was made for multiple comparisons. **d** Polarization indices for chromosomes with or without nucleolar or lamina association (with compartment B) in proerythroblasts. $\overline{LA}$ and $\overline{NA}$: with neither lamina nor nucleolar association (n = 1277 chromosomes, median = 0.59, 25% quantile = 0.48, 75% quantile = 0.72, minimum = 0.09, maximum = 1). $\overline{LA}$ and NA: without lamina association but with nucleolar association (n = 465 chromosomes, median = 0.65, 25% quantile = 0.53, 75% quantile = 0.78, minimum = 0.06, maximum = 1). LA and $\overline{NA}$: with lamina association but without nucleolar association (n = 2267 chromosomes, median = 0.57, 25% quantile = 0.45, 75% quantile = 0.71, minimum = 0.02, maximum = 1). LA and NA: with both lamina and nucleolar associations (n = 821 chromosomes, median = 0.62, 25% quantile = 0.49, 75% quantile = 0.74, minimum = 0.03, maximum = 1). Dots, lines and boxes are defined as in Fig. 3e. Source data are provided as a Source data file.

whereas compartment B scores were positively correlated with chromosome surface ratios (Fig. 6f), inconsistent with our experimental observations. These results suggest that intrachromosomal A–A and B–B interactions can jointly establish the polarized organization of the A/B compartments. However, contrary to our observations, the TADs with the strongest A and B identities tend to serve as central hubs of A–A and B–B interactions and tend to localize towards the centers of the A and B compartments, if only A–A and B–B interactions are considered. Finally, we added to the model extra-chromosomal interactions between TADs and nuclear contents surrounding the chromosome: Localization of a TAD to the chromosome surface causes an energy change dependent on the TAD's compartment score. The interaction between compartment A TADs and their chromosome surroundings, termed A–S interactions, and the similarly defined B–S interactions were set to be governed by two independent energy parameters. By tuning the strengths of the A–S interactions and B–S interactions, we obtained a positive correlation between the compartment A scores and chromosome surface ratios, and a moderately negative correlation between the compartment B scores and chromosome surface ratios, all while

maintaining the polarized organization of the A/B compartments (Fig. 6g–i). This last setting qualitatively recapitulated our experimental observations. These results suggest that a balance between the extra-chromosomal and intra-chromosomal interactions is necessary to establish and maintain the observed chromatin organization.

## Discussion

In this paper, we report an imaging method, termed MINA, that provides a multiscale and multi-faceted picture of chromatin folding and nucleome architectures in individual cells in complex mammalian tissue, allowing the mapping of cell-type-specific chromatin structural features related to gene expression as well as co-variation of different nucleome architectures. We expect this imaging method to be broadly applicable to mapping nucleome architectures in other mammalian tissue types, throughout development and aging, and in health and disease conditions.

Using MINA, we observed three facets of chromatin arrangement that vary among cell types: (1) In terms of fine-scale chromatin folding, we identified a promoter-enhancer interaction in

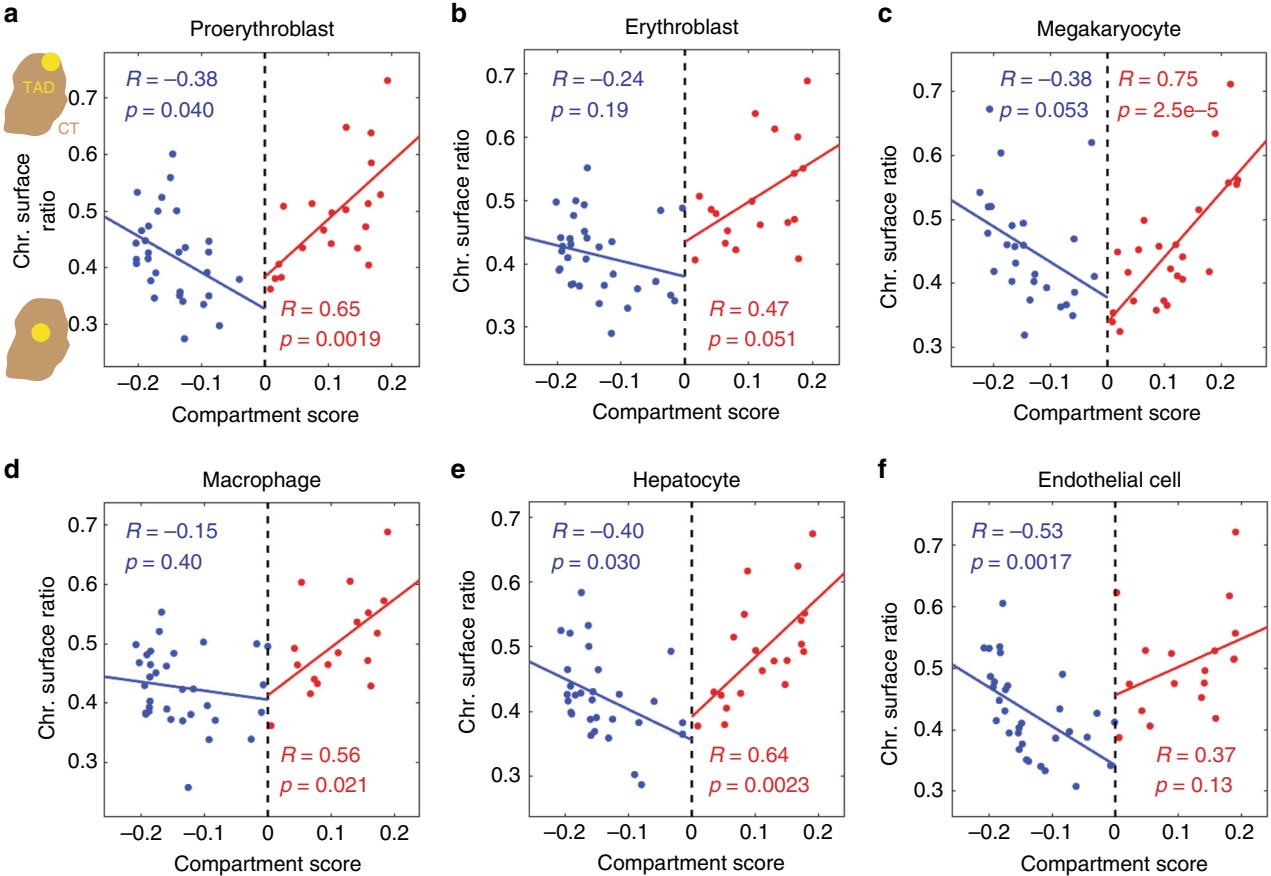

**Fig. 5 Chromosome surface ratios of TADs versus their compartment scores in different cell types. a** Proerythroblast. The cartoons along the *y*-axis of A illustrate TADs localizing to the surface or interior of the chromosome territory (CT). **b** Erythroblast. **c** Megakaryocyte. **d** Macrophage. **e** Hepatocyte. **f** Endothelial cell. Each dot represents a TAD. The lines show linear regression fits. Correlation coefficients (R) and the corresponding p values were calculated for compartment-B TADs (blue) and compartment-A TADs (red) separately. The *p* values were calculated for Pearson's correlation using a two-sided Student's *t* distribution. No adjustment was made for multiple comparisons. Source data are provided as a Source data file.

the cis-regulatory region of *Scd2* that was enriched in fetal liver hepatocytes (Fig. 2d–f). (2) At a larger scale, we found that the folding schemes of TADs to A/B compartments also differ among fetal liver cell types, with some of the same TADs on the genomic map spatially assigned to different compartments depending on the cell type (Fig. 3b and Supplementary Fig. 2). (3) In terms of the positioning of chromatin relative to other nuclear structures, we measured the spatial proximity of chromatin regions to nuclear lamina and nucleoli, and found that the lamina/nucleolar association ratios can systematically vary between cell types, e.g. between the closely related erythroblasts and proerythroblasts, independent of the compartmentalization differences (Fig. 4a, b). Our observations of cell-type-specific enrichment of *Scd2* promoter-enhancer interaction, different Chr19 compartmentalization schemes, and their correlation with transcript levels in mouse fetal liver suggest a model wherein local promoter-enhancer interactions and global compartmentalization of TADs are each associated with and may jointly regulate transcription activities of genes, and in turn define cell types and functions in mammalian tissue. Although TADs may switch compartments in different cell types, we also observed several general principles of TAD and compartment organization regardless of cell type in fetal liver, including polarized organization of the A/B compartments, and chromosome-surface localization of TADs with strong compartment A/B epigenetic identities. The fact that these principles are generally conserved in different cell types suggests that they are

achieved through specific mechanisms and are functionally important. Our polymer simulation suggests that four types of interactions are involved in the establishment and maintenance of the observed organization principles: A–A, B–B, A–S, and B–S interactions. Among these, the intra-chromosomal A–A and B–B interactions could be driven by the self-associations (through phase-separation or other mechanisms) of active and inactive epigenetic marks and proteins on chromatin. The recently observed liquid droplet formation by RNA polymerase II, mediator, BRD4, and HP1α may play significant roles for the self-associations[34–38]. The extra-chromosomal B–S interaction may be attributed to the observed and apparently regulated association of compartment B regions with nuclear lamina and nucleoli. Particularly, recent studies have shown that the interactions between heterochromatin and nuclear lamina can drastically change global chromatin organization[39,40]. The extra-chromosomal A–S interaction may be attributed to the localization of gene-dense regions and actively transcribing genes to the chromosome surface[41,42]. A–S and B–S interactions may also be inter-chromosomal interactions[43]. Further investigations are required to pinpoint the major sources of these interactions. Tuning the balance among the four types of interactions could lead to alternative chromosome architectures, such as radially organized A/B compartments (Fig. 6a, b), similar to the chromatin organization observed in senescence-associated heterochromatic foci[44] and consistent with the results from a recent simulation of senescent cells[40]. Together,

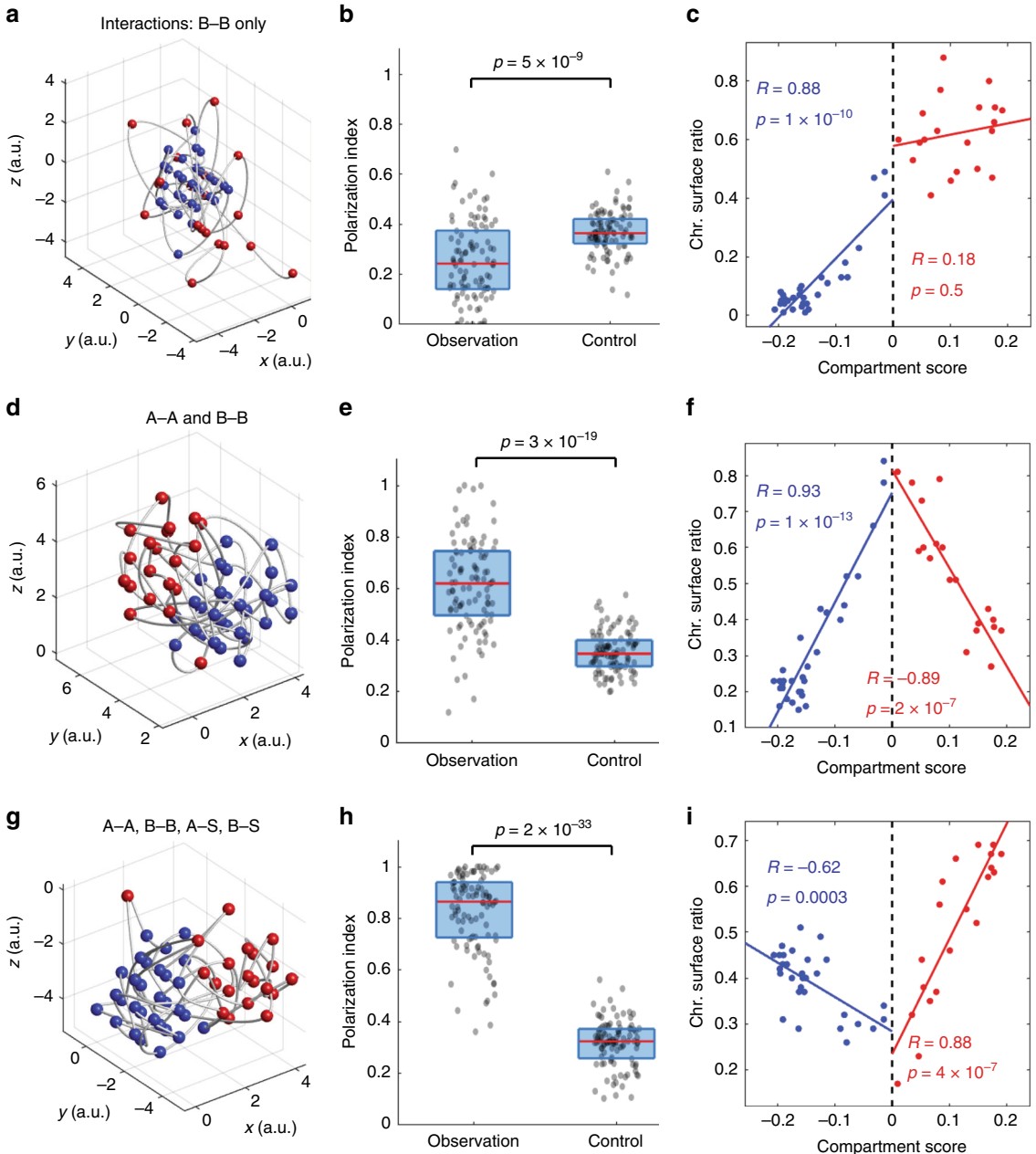

**Fig. 6 Computer simulations of a polymer model showing that both intra- and extra-chromosomal interactions are required to establish the observed features of chromatin organization.** The simulated polymer contained 50 monomers representing the 50 TADs in Chr19. The monomers were assigned epigenetic scores based on the compartment scores of TADs in hepatocytes. **a–c** Simulations with only intra-chromosomal interactions between compartment-B TADs (B–B interaction). **d–f** Simulations with interactions between compartment-B TADs and interactions between compartment-A TADs (A–A interaction). **g–i** Simulations with A–A and B–B interactions as well as additional interactions of A/B TADs with extra-chromosomal components when the TADs were located at the chromosome surface (A–S and B–S interactions). **a, d, g** Traces of individual polymer conformations generated by the simulations. Red: compartment-A TADs. Blue: compartment-B TADs. **b, e, h** Polarization indices of the simulated chromosomes and the randomization controls. Dots, lines and boxes are defined as in Fig. 3e. Observation medians = 0.24 (**b**), 0.63 (**e**), 0.87 (**h**). Control medians = 0.36 (**b**), 0.39 (**e**), 0.32 (**h**). Data from 100 simulated chromosomes were used to generate each observation and control group in (**b, e, h**). **c, f, i** Chromosome surface ratios of TADs in the simulated chromosomes versus the compartment scores. Correlation coefficients were calculated for compartment-B TADs (blue) and compartment-A TADs (red) separately. The $p$ values were calculated for Pearson's correlation using a two-sided Student's t distribution. No adjustment was made for multiple comparisons. Source data are provided as a Source data file.

these results suggest that genome structures across multiple length scales, in conjunction with other nuclear structures, may regulate gene activity and cell functions, and could be altered through intra- and extra-chromosomal interactions to define different cell states and cell types.

## Methods

**Probe design.** The complex pools of primary oligonucleotide probes were enzymatically amplified from template oligo pools generated with array-based oligo pool synthesis[45–47]. The template oligo pool for tracing 50 TADs of mouse Chr19 was designed as follows: Each oligo in this pool consisted of four regions: from 5′ to 3′, (i) a 20-nucleotide (nt) forward priming region, (ii) a 30-nt readout region,

(iii) a 30-nt targeting region, and (iv) a 20-nt reverse priming region. The forward and reverse priming regions help to selectively amplify this oligo pool in a limited-cycle PCR procedure (described below). The readout region and the targeting region bind dye-labeled secondary probes and the genomic target, respectively. The sequences of the 20-nt priming regions were generated from random sequences[29] that were screened to ensure the lack of significant homology with the mouse genome and good performance in PCR priming. The sequences of the 30-nt readout regions were generated from concatenations of the 20-nt sequences with 10-nt halves of other sequences created as above. The 30-nt targeting regions were chosen from the genomic sequence of the center 100-kb of each TAD[9] with the software OligoArray2.1[48], with the following constraints: The melting temperatures of the chosen sequences are no less than 60 °C; the melting temperatures of potential cross hybridization among the sequences do not exceed 70 °C; the melting temperatures of potential secondary structures in the sequences do not exceed 70 °C; the GC contents of the sequences are between 30% and 90%; there is no consecutive repeat of seven or more A's, T's, G's or C's; and the chosen sequences do not overlap each other. The chosen sequences were further screened against the mouse genome and transcriptome with BLAST+[49] to ensure they appear only once in the genome and do not overlap any transcribed regions. The genomic positions of the TADs were downloaded from http://chromosome.sdsc.edu/mouse/hi–c/download.html. The coordinates of the targeted genomic regions are listed in Supplementary Data 1.

The template oligo pool for tracing 19 5-kb regions upstream of *Scd2* was designed with the following modifications: Each template oligo contained two identical 30-nt readout regions flanking the targeting region to double the fluorescent signal from secondary probes; the melting temperatures of the chosen targeting sequences are no less than 66 °C; the melting temperatures of potential cross hybridization among the targeting sequences do not exceed 72 °C; the melting temperatures of potential secondary structures in the targeting sequences do not exceed 76 °C; there is no consecutive repeat of six or more identical nucleotides in the targeting region; targeting sequences are allowed to overlap each other by at most 20 nucleotides. These modifications helped to increase the probe density and compensate for the shorter lengths of the targeted genomic regions. The genomic coordinates of the targeted regions are listed in Supplementary Data 2.

Probe design for RNA MERFISH was similar to that previously described[29]. Each oligo in the RNA MERIFSH template oligo pool contained six regions: (from 5′ to 3′) (i) a 20-nt forward priming region, (ii) a 20-nt readout region, (iii) a 30-nt targeting region, (iv) a second 20-nt readout region, (v) a third 20-nt readout region, and (vi) a 20-nt reverse priming region. The sequences of the priming regions were created as above. For the 20-nt readout regions, we used 16 fast-binding readout sequences introduced in a previous MERFISH study[50]. Each RNA species was assigned a unique combination of 4 out of the 16 readout sequences as a unique barcode. Each template oligo only includes 3 out of the 4 readout sequences due to oligo synthesis length limit. In all, 48 template oligos were designed to target each RNA species of interest. The 48 template oligos rotate to carry different combinations of 3 out of the 4 readout sequences, so that each readout sequence is included 36 times among the 48 oligos. For the combinatorial barcoding, we used the previously reported Modified Hamming Distance 4 (MHD4) code for MERFISH[29]. The codes assigned to individual genes are listed in Supplementary Data 3. The targeting regions were designed from the transcript sequences of 55 cell type marker genes based on previous single-cell RNA sequencing studies[31], and 82 additional genes located on mouse Chr19. The design criteria for the targeting regions are: The melting temperatures of the targeting sequences are no less than 66 °C; the melting temperatures of potential cross hybridization among the targeting sequences do not exceed 72 °C; the melting temperatures of potential secondary structures in the targeting sequences do not exceed 76 °C; there is no consecutive repeat of six or more identical nucleotides in the targeting region; targeting sequences are not allowed to overlap each other. The targeting sequences were screened against the genome and transcriptome to ensure that each sequence is unique in the genome and is transcribed from only one gene. The template oligo sequences for all three oligo pools introduced above are listed in Supplementary Data 4.

**Probe synthesis**. The designed template oligo pools were ordered from CustomArray, GenScript. To synthesize primary probe sets from the template oligo pools, we followed a procedure that involved limited-cycle PCR, in vitro transcription, reverse transcription, alkaline hydrolysis and purification[26,29,50]. All PCR primers and reverse transcription primers used in probe synthesis, as well as dye-labeled secondary probes used in the sequential hybridization and imaging procedure below, were purchased from Integrated DNA Technology (IDT), Inc. The sequences of the primers and secondary probes are listed in Supplementary Data 5 and 6.

**Sample preparation**. *Animals*: Pregnant female C57BL/6 mice at the age of 8–15 weeks from the Jackson Laboratory were used for all experiments. All mice were maintained under 12 h light/12 h darkness cycles with constant conditions of temperature (22 °C) and humidity (40–60%). All procedures have been approved by the Institutional Animal Care and Use Committee of Yale University.

*Oligo conjugation to WGA*: We adapted a previously published strategy for antibody-oligo conjugation[51,52] to conjugate DNA oligonucleotides to WGA. First,

DBCO-PEG5-NHS ester (Kerafast) was diluted to a concentration of 10 mM in anhydrous dimethyl sulfoxide (DMSO). In all, 2.7 μL of the solution was added to 100 μL of 2 mg/mL WGA in Dulbecco's phosphate-buffered saline (DPBS; Gibco, 14190144) and incubated at room temperature for 1 h. We then terminated the reaction using a spin column-based dialysis membrane (Amicon, 10 kDa molecular weight cut off). Next, 20 μL of 100 μM 3′-azide-modified oligonucleotide were combined with the purified DBCO-labeled WGA. The reaction was incubated at 4 °C for at least 12 h to obtain the oligo-conjugated WGA. The sequence of the azide-modified oligo is: CGGTACGCACTTCCGTCGACGCAATAGCTC/3AzideN/. The corresponding ATTO565-labeled secondary probe has the following complementary sequence: /5ATTO565N/AGAGCTATTGCGTCGACGGAAGTGCGTACCG. Both oligos were ordered from Integrated DNA Technology (IDT), Inc.

*Tissue sectioning*: Pregnant females were euthanized by isoflurane inhalation and cervical dislocation. Embryos at E14.5 were dissected from the uterus and immersed in ice-cold DPBS. Fetal liver was dissected from the embryo and embedded in a 25-mm × 20-mm × 5-mm Tissue-Tek Cryomold (VWR, 25608-916) with optimal cutting temperature compound (Tissue-Tek O.C.T.; VWR, 25608-930). Frozen tissue blocks were stored at −80 °C until cryosectioning. Prior to cryosectioning, 40-mm-diameter #1.5 glass coverslips were treated with 0.01% poly-L-lysine (Millipore, A-005-C) at room temperature for 15 min. Frozen E14.5 fetal liver tissue block was cryosectioned at −15 °C at a thickness of 10 μm. Tissue sections were immediately fixed in 4% formaldehyde (PFA) in DPBS for 20 min at room temperature, and washed with DPBS for 3 min twice. We have previously shown that this osmotically balanced fixation condition does not lead to detectable chromosome shrinkage during fixation[53]. Next the tissue sections were washed with 1× Hanks' balanced salt solution (HBSS) once for 3 min, stained with oligo-conjugated WGA at a concentration of 2–5 μg/mL in 1× HBSS with 2000× diluted murine RNase inhibitor (New England Biolabs, M0314L) at 37 °C for 20 min, and washed with 1× HBSS for three times. The samples were then post-fixed in 4% PFA for 10 min, washed with DPBS twice, permeabilized with 0.5% v/v Triton X-100 (Sigma, T8787) in DPBS for 15 min at room temperature, and washed twice with DPBS. Afterwards, the sections were directly used for MINA primary probe hybridization.

**MINA primary probe hybridization**. Tissue sections were blocked with blocking buffer (1% w/v bovine serum albumin (BSA), 22.52 mg/mL glycine (AmericanBIO, 56-40-6), 0.1% v/v Tween 20 in DPBS) with 1000× diluted murine RNase inhibitor at room temperature for 30 min. The sample was then incubated with anti-fibrillarin primary antibody (Abcam Cat# ab5821) at a concentration of 1:100 in blocking buffer with 100× diluted murine RNase inhibitor at 4 °C overnight and washed 3 times with DPBS for 5 min each. Tissue sections were then incubated with Alexa Fluor 647-conjugated anti-rabbit secondary antibody (Molecular Probes Cat# A-31573) at a concentration of 1:1000 in blocking buffer with 1000× diluted murine RNase inhibitor at room temperature for 1 h and washed three times with DPBS for 5 min each. The sample was then post-fixed in 4% PFA for 10 min, washed with DPBS twice, treated with 0.1 M HCl for 5 min and washed with DPBS twice. Tissue sections were then incubated in pre-hybridization buffer composed of 50% formamide and 2 mM Ribonucleoside vanadyl complexes (Sigma-Aldrich, R3380) in 2× saline-sodium citrate (SSC) buffer for 5 min at room temperature. Hybridization buffer comprising 50% formamide, 0.1% wt/yeast tRNA (Life Technologies, 15401011), 10% dextran sulfate (Sigma, D8906-50G) and 100× diluted murine RNase inhibitor in 2× SSC was prepared. For heat denaturation, 20ul hybridization buffer were dropped onto a glass slide. The coverslip with tissue sections was flipped and placed on top of the glass slide so that the tissue section was in contact with the hybridization buffer. The coverslip-slide assembly was placed on top of an 80 °C heat block for 3 min. The coverslip was then removed and briefly washed with 2× SSC. Next, for probe hybridization, 12.5 μL of hybridization buffer with 24–28 μM MERFISH probes and chromatin tracing probe sets targeting 50 TADs of Chr19 and 19 loci upstream of *Scd2* at a concentration of 8 and 4 μM, respectively, were dropped onto a piece of parafilm. Note that we applied the primary FISH probes after the heat denaturation. This procedure avoided heat denaturation of the RNase inhibitor in the hybridization buffer and allowed multiplexed DNA and RNA FISH to be performed simultaneously. The coverslip was then flipped and placed onto the parafilm so that the tissue section was in contact with the hybridization buffer containing probes. The assembly was incubated for 24–28 h at 37 °C in a humid chamber. The tissue sections were then washed twice with 0.1% v/v Tween 20 in 2×SSC at 60 °C for 15 min each, and once more at room temperature for 15 min. Previous electron microscopy and super-resolution imaging studies have shown that these denaturation and hybridization conditions largely preserve the chromatin ultrastructure[54–56]. Finally, we applied 0.1-μm yellow-green beads (Invitrogen, F8803) resuspended in 2× SSC to the sample so that the beads attached to the coverslip and could serve as fiducial markers for the correction of sample drift during sequential hybridization rounds.

**MINA imaging**. After the primary probe hybridization, the sample was repeatedly hybridized with different secondary probes, imaged, and photobleached. For MERFISH measurements, we used Alexa Fluor 750-conjugated 20-nt secondary probes with sequences complementary to the readout regions on the MERFISH primary probes. For chromatin tracing, we used Alexa Fluor 647 and ATTO 565-conjugated 30-nt secondary probes with sequences complementary to the readout

regions on the chromatin tracing primary probes. All dyes were attached to the 5' ends of the secondary probes (Supplementary Data 6).

To automatically perform buffer exchange during the multiple rounds of secondary hybridization, we used a Bioptech's FCS2 flow chamber and a computer-controlled, home-built fluidics system[26,29]. For each round of secondary hybridization, the sample was first treated with secondary hybridization buffer (20% v/v ethylene carbonate (Sigma-Aldrich, E26258) and 0.05% murine RNase inhibitor in 2× SSC) containing 3.75 nM of the corresponding MERFISH secondary probe and 7.5 nM each of the corresponding Alexa Fluor 647 and ATTO 565 labeled chromatin tracing secondary probes, and was incubated at room temperature for 20 min. We then sequentially flowed through the chamber 2 mL of readout wash buffer (20% v/v ethylene carbonate in 2× SSC) and 2 mL of imaging buffer with an oxygen scavenging system[57] (50 mM Tris-HCl pH 8.0, 10% wt/v glucose, 2 mM Trolox (Sigma-Aldrich, 238813), 0.5 mg/mL glucose oxidase (Sigma-Aldrich, G2133), 40 μg/mL catalase (Sigma-Aldrich, C30), 0.05% murine RNase inhibitor in 2× SSC). The imaging buffer was stored under a layer of mineral oil (Sigma, 330779) to prevent continuous oxidation. Next, the sample was imaged at multiple fields of view. At each field of view, we took z-stack images with 750-nm, 647-nm, 560-nm, and 488-nm laser illuminations. The z-stacks had a step size of 200 nm, and an exposure time of 0.4 s at each step. The ranges of the z-stacks are 7 μm in z. After the imaging, we switched sample buffer to 2× SSC, and photobleached the sample by simultaneous illumination with the 750-nm, 647-nm, and 560-nm lasers for 25 s.

We performed 40 rounds of secondary hybridization in total, denoted as hyb 1-40, as well as a round of pre-hybridization, imaging and photobleaching before the 40 rounds of secondary hybridization, denoted as pre-hyb. Prior to sample assembly into the flow chamber, in pre-hyb, we hybridized to the sample the first MERFISH secondary probe in secondary hybridization buffer at room temperature for 20 min and washed twice with 2× SSC. The sample was then mounted onto the flow chamber and microscope. Next, in the imaging step of pre-hyb, the first round of MERFISH images and 3D fibrillarin antibody staining images were sequentially collected with z-stepping in the 750-nm channel and 647-nm channel and photobleached as in the other rounds of secondary hybridization. The 16 rounds of MERFISH readout hybridization were imaged in the 750-nm channel from pre-hyb to hyb 15. The first 40 TADs of Chr19 were imaged in the 647-nm channel from hyb 1 to hyb 40; the last 10 TADs of Chr19 were collected in the 560-nm channel from hyb 21 to hyb 30. The 19 consecutive loci upstream of Scd2 were imaged in the 560-nm channel from hyb 1 to hyb 19. After hyb 40, we flowed through the chamber 4 mL of SYTOX Deep Red Nucleic Acid Stain (Invitrogen, S11380) at a 1:2000 concentration in DPBS with 0.1% v/v Triton X-100, or 4 mL diamidino-2-phenylindole (DAPI; Thermo Scientific, 62248) in DPBS at a 1:1000 concentration, and incubated the sample for 30 min at room temperature. We then flowed DPBS and imaging buffer through the chamber and imaged SYTOX nucleus staining in the 560-nm channel or DAPI nucleus staining in the 405-nm channel with z-stepping as in previous imaging steps.

**Imaging system**. For imaging, we used a home-built microscope with a Nikon Ti2-U body, a Nikon CFI Plan Apo Lambda 60× Oil (NA1.40) objective lens, and an active auto-focusing system[58]. A 750-nm laser (2RU-VFL-P-500-750-B1R, MPB Communications) was used to excite and image Alexa Fluor 750 on secondary probes. A 647-nm laser (2RU-VFL-P-1000-647-B1R, MPB Communications) was used to excite and image Alexa Fluor 647 on secondary probes and on the anti-rabbit secondary antibody. A 560-nm laser (2RU-VFL-P-1000-560-B1R, MPB Communications) was used to excite and image ATTO 565 on secondary probes and the SYTOX nucleic acid stain. A 488-nm laser (2RU-VFL-P-500-488-B1R, MPB Communications) was used to excite and image the yellow-green fluorescent beads for drift correction. A 405-nm laser (OBIS 405 nm LX 50 mW, Coherent) was used to excite and image the DAPI stain. The five laser lines were directed to the sample using a multi-band dichroic mirror (ZT405/488/561/647/752rpc-UF2, Chroma) on the excitation path. On the emission path, we had a multi-band emission filter (ZET405/488/561/647-656/752 m, Chroma) and a Hamamatsu Orca Flash 4.0 V3 camera. The pixel size of our system was 107.9 nm. The color shift between channels was canceled by taking z-stack calibration images of 100-nm Tetraspeck beads (Invitrogen) attached to a coverslip surface[26]. To automatically scan and image multiple fields of view, we used a computer-controlled motorized x–y sample stage (SCAN IM 112×74, Marzhauser).

**Data analysis**. *RNA MERFISH analysis*: all analyses in this work were performed with MATLAB (MATLAB) version R2018a. To efficiently analyze MERFISH images, we implemented a pixel-based MERFISH analysis pipeline similar to that introduced in a previous report[50], with modifications. First, to correct for sample drift between the different rounds of MERFISH imaging, we fitted the fiducial bead markers with 2D Gaussian functions to determine the movement of their center positions in x and y, and subtracted this movement from the MERFISH images with image translation. Next, for each drift-corrected raw RNA image, we derived a background image by image opening with a disk-shaped morphological structuring element with a radius of 5 pixels. We subtracted the background image from the RNA image, and identified regional maxima in this new image. All regional maxima with pixel intensities higher than a threshold were identified as potential RNA signals. We then generated a binarized RNA image where all RNA-occupied pixels (the subset of regional maxima pixels with high enough intensities) are ones

and all other pixels are zeros. We further dilated the binarized image with a square-shaped morphological structuring element with a width of 3 pixels to enlarge the area occupied by each RNA molecule (to account for the possibility that one RNA molecule may occupy more than one pixels). We grouped all 16 binarized images from the 16 rounds of MERFISH readout imaging at each height of the z-stack, and determined whether the 1/0 values of each pixel across the 16 images fitted one of the codes in the MERFISH codebook. If so, the pixel was determined to contain one molecule of the corresponding RNA. All adjacent pixels determined to contain the same RNA species were counted as one molecule of that RNA species. To determine the threshold value for the identification of potential RNA signals, we used an adaptive procedure to screen multiple threshold values for each round of imaging, so that the relative abundance of total RNA molecules in different rounds of imaging fitted the expected relative abundance based on the bulk RNA-seq data and the MERFISH codebook, also the final molecule counts for different RNA species were best correlated with the bulk RNA-seq data.

*Cell segmentation*: because our tissue section largely consisted of a monolayer of cells, we segmented these cells in 2D based on the WGA labeling pattern. We first averaged the WGA images in each z-stack along the z direction, and normalized the average image so that the minimum pixel intensity is zero and the maximum pixel intensity is 1. We then calculated the background profile of each averaged WGA image using the adaptthresh function in MATLAB with a sensitivity of 0.1 and a neighborhood size of 41 pixels. We divided the averaged WGA image by this background profile to remove the background, and then thresholded the image so that the 1st and 99th percentiles of the pixel intensities were set as the minimum and maximum intensities, respectively. Next, we applied an image closing procedure with a disk-shaped morphological structuring element with a radius of 15 pixels to get more connected WGA boundaries. Finally, we applied the watershed algorithm to the closed image to segment individual cells.

*Cell type clustering and identification*: to count single-cell RNA copy numbers, we first corrected for sample drift between the WGA and RNA images with the fiducial bead markers as above. We then identified the area of each single cell from the cell segmentation result, and shrank the area by erosion with a disk-shaped morphological structuring element with a radius of 3 pixels. This erosion procedure reduced the chance of mis-assignment of RNA molecules between neighboring cells. Next, we counted the single-cell copy numbers of each probed RNA species in the cell area. To cluster cells based on the RNA copy number profiles, we applied the Louvain-Jaccard clustering algorithm[59,60], and visualized all the clusters with t-distributed stochastic neighbor embedding (t-SNE)[61]. Both algorithms were previously used to analyze single-cell RNA sequencing data[62]. Only the 55 marker genes were included in the Louvain-Jaccard clustering and t-SNE analyses. We determined the cell type identities of the clusters based on the enrichment of marker gene transcripts. Some of the clusters automatically identified by the Louvain-Jaccard algorithm had similar marker gene expression profiles and were located next to each other on the t-SNE plot. We manually merged these clusters and regarded them as the same cell type, as in previous report of single-cell RNA sequencing analysis[62].

*Determination of 3D Chromatin traces*: the 3D chromatin traces were determined from the sequential DNA FISH images with a previously reported analysis procedure[26], with minor modifications. First, we fitted z-stack images of probed DNA loci to 3D Gaussian functions to determine their center positions in x, y, and z. To subtract sample drift from the DNA loci positions, we determined the 3D positions of the fiducial beads in the same imaging rounds with the same fitting algorithm, and subtracted the bead movement. The drift-corrected DNA loci positions were linked into traces based on the spatial clustering of the positions from different rounds of imaging. The traces were used to calculate the mean spatial distances between pairs of DNA loci.

*Determination of Hi–C contact frequency between TADs*: Hi–C data of E14.5 mouse fetal liver were downloaded from Gene Expression Omnibus GSE70181[32]. Hi–C contact frequencies between TADs were calculated as in our previous report[26]. In brief, we first summed all counts of contact between each pair of TADs. We then divided the total count by the product of the genomic lengths of the two TADs. This normalization of the total count to the genomic sizes of the two TADs yields the contact frequency.

*Identification of compartment assignment of TADs*: the A/B compartmentalization of TADs was determined with a computational procedure introduced in our previous work[26]. This procedure was an adaptation of the computational workflow to identify A/B compartments from Hi–C data[8]. Briefly, we started from the mean spatial distance matrix of the chromosome, in which individual entries were the mean spatial distances between pairs of TADs. The matrix showed two general features of chromatin organization: (1) Spatial distance between TADs generally increases with increasing genomic distance, shown by shorter mean spatial distances near the diagonal line. (2) There are deviations from the first feature due to long-range interactions/repulsions (compartmentalization). To cancel the contribution from genomic distance, we fitted a power-law function to the plot of mean spatial distance versus genomic distance. The function generated the expected spatial distance at each genomic distance. We then normalized the mean spatial distances between TADs to their corresponding expected spatial distances, and obtained a normalized distance matrix. This matrix shows how the observed spatial organization deviates from the power-law scaling. Next, we calculated the Pearson correlation coefficient between each pair of rows or columns of the normalized matrix. The Pearson correlation matrix further highlighted the TADs' spatial

organization due to compartmentalization. Finally, we applied a principal component analysis to the Pearson correlation matrix, treating each column as a variable and each row as an observation, or vice versa. The coefficients of the first principal component were the compartment scores, with positive and negative values indicating the two compartments, respectively. Because the direction of the principal vector can be arbitrarily flipped, we require compartment scores to be positively correlated with gene density, based on previous Hi–C studies[63]. Under this convention, compartment-A regions have positive scores and compartment-B regions have negative scores.

*Quantification of the polarized arrangement of A/B compartments*: we used a previously established metric termed polarization index to quantify the extent of the polarized organization of compartments A and B[26]. Briefly, in each imaged copy of chromosome, we built a 3D convex hull for all TADs associated with compartment A with MATLAB function convhull, and similarly built a 3D convex hull for all TADs associated with compartment B. We calculated the volumes of the two hulls, denoted by $V_A$ and $V_B$ respectively, and the shared volume of any spatial overlap between the two, denoted by $V_S$ (if there is no overlap, $V_S$ equals zero). The polarization index is calculated as:

$$PI = \sqrt{\left(1 - \frac{V_S}{V_A}\right)\left(1 - \frac{V_S}{V_B}\right)}, \quad (1)$$

or the geometric mean of the non-shared proportions of compartments A and B. Under this definition, a polarization index of one indicates that the two compartments are completely polarized in space. A polarization index of zero indicates that the two compartments are completely overlapping, or one wraps around the other. We used two-sided Wilcoxon rank sum test to compare observed polarization indices with the polarization indices of the randomization controls (where we randomized the compartment assignments of TADs) because Wilcoxon rank sum test does not require the populations to be normally distributed.

*Determination of lamina association of TADs*: to determine if TADs were associated with nuclear lamina, we identified spatial voxels near the edges of the imaged nuclei to approximate nuclear regions adjacent to the lamina: For each image in a DAPI/SYTOX fluorescent z-stack, we first normalized the image to its maximum pixel intensity. We then calculated the background profile of the normalized image using the adaptthresh function in MATLAB with a sensitivity of 0.5. We divided the normalized nuclear image by this background profile to remove the background, and then thresholded the image so that the 1st and the 3rd quartiles of the pixel intensities were set as the minimum and maximum intensities, respectively. Next, we applied an image opening procedure by reconstruction with a disk-shaped morphological structuring element with a radius of 25 pixels to reduce image noise. We calculated the 2D gradient profile of the resulting image to highlight the nuclear edges, applied a Gaussian filter to the gradient image with a standard deviation of 5 pixels to clean up the image, and binarized the filtered gradient image using the imbinarize function in MATLAB with the "adaptive" option and a sensitivity of 0.1. We considered a TAD as being associated with nuclear lamina if the TAD's distance to a nearest nuclear edge voxel in the z-stack is <200 nm. In Fig. 4d and Supplementary Fig. 7, we consider the compartment B of a chromosome as being associated with nuclear lamina if any of the compartment B TADs in this chromosome is associated with nuclear lamina.

*Determination of nucleolar association of TADs*: to determine if a TAD was associated with nucleolus, we identified nucleolus-occupied spatial voxels by binarizing the z-stack images of nucleoli. We first determined an adaptive threshold profile for the binarization using the following procedure: For each z-stack, we generated a maximum projection along the z direction, and filtered the maximum-projection image with a median filter using the medfilt2 function in MATLAB; we then normalized the filtered image by its maximum pixel intensity, which later was also used as a normalization factor for the whole z-stack; we used the normalized image to calculate an adaptive threshold profile with the MATLAB adaptthresh function. Next, we binarized individual images in the z stack by first applying a median filter to each image, then normalizing each filtered image with the normalization factor above, and finally binarizing each normalized image based on the adaptive threshold profile. After the binarization, a voxel occupied by nucleolus has value 1, and a voxel not occupied by nucleolus has value 0. We considered a TAD as being associated with nucleolus if the TAD's distance to the nearest nucleolus-occupied voxel is <200 nm. In Fig. 4d and Supplementary Fig. 7, we consider the compartment B of a chromosome as being associated with nucleolus if any of the compartment B TADs in this chromosome is associated with nucleolus.

*Determination of chromosome-surface localization of TADs*: to determine the chromosome-surface localization of a TAD, we constructed a 3D convex hull for all imaged TADs of the chromosome and determined if the TAD in question is on the surface of the 3D convex hull. TADs on the surface of the 3D convex hull were considered as being localized to the chromosome surface.

*Sample sizes, replicates, and exclusions*: our sample sizes for different cell types are: Proerythroblast: $N = 4873$. Megakaryocyte: $N = 358$. Macrophage: $N = 1773$. Hepatocyte: $N = 7543$. Erythroblast: $N = 8525$. Endothelial Cell: $N = 911$. Other: $N = 1181$. These measurements were from four biological replicates. All analyses were performed using the same overall population of cells. When analyzing single-cell RNA data, we excluded "cells" larger than 20,000 pixels in area as these were usually empty regions in tissue sections. We excluded "cells" smaller than

2500 pixels in area as these were usually non-cell particles. We excluded partial cells overlapping the edges of each field of view. We excluded cells with <10 detected RNA molecules to ensure high quality in the cell type identification analyses.

**Monte Carlo simulation of chromosome conformation with a lattice polymer model**. To simulate the compartmentalization of TADs in a chromosome, we built a minimal polymer model on a cubic lattice in 3D. The chromosome was modeled as a linear, self-avoiding polymer composed of 50 monomers. The monomers could only occupy discrete positions on the lattice. The distance between adjacent monomers along the polymer were constrained during the entire simulation, and must be no smaller than 1 and no bigger than 4 (the units of all distances mentioned in this section are the lattice units). In each simulation, we started with a random initial polymer conformation, and simulated a set of monomer moves with a Monte Carlo procedure. For each move, we first randomly chose one monomer, and randomly chose one of the six closest positions next to the monomer on the lattice (plus or minus 1 in x, y, or z) as the attempted new position for this monomer. We then determined whether the new position was already occupied by another monomer, and whether the move would violate the distance constraint for adjacent monomers. If either was the case, we gave up this move attempt, and restarted another attempt from the step of randomly choosing a monomer. If neither was the case, we calculated the energies of the polymer conformations with ($E_{new}$) or without ($E_{now}$) the attempted move, and accepted or rejected the move using the Metropolis algorithm: We generated a uniformly distributed random number p in the interval (0, 1). If p<exp ($E_{now} - E_{new}$), we accepted the move; otherwise, we rejected the move. If the move was rejected, we reinitiated another move attempt from the step of randomly choosing a monomer. This procedure was repeated until an attempted move was accepted. We simulated 60,000 accepted moves in each run, which is sufficient to reach equilibrium as the mean energy of the population of simulated conformations showed no statistically significant changes in the last 10,000 moves. And we sampled 100 polymer conformations with 100 independent runs for each set of energy parameters.

To calculate the energy of a polymer conformation, we regarded the 50 monomers as the 50 probed TADs in Chr19, and assigned the measured compartment scores from fetal liver hepatocytes to the monomers. The energy was calculated with the following function:

$$E = \sum_{ij} g_{AA}S_iS_j + \sum_{pq} g_{BB}S_pS_q + \sum_m g_{AS}(S_m - \overline{S_A}) + \sum_n g_{BS}(\overline{S_B} - S_n), \quad (2)$$

where ij denotes all pairs of compartment-A TADs with a distance closer than 2, which are considered interacting. $g_{AA}$ is the energy parameter for A–A interactions. $S_i$ and $S_j$ are the compartment-A scores of the two interacting TADs. pq denotes all pairs of compartment-B TADs with a distance closer than 2. $g_{BB}$ is the energy parameter for B–B interactions. $S_p$ and $S_q$ are the compartment-B scores of the two interacting TADs. m denotes all compartment-A TADs that are located on the surface of the polymer conformation. $g_{AS}$ is the energy parameter for A-surface interactions. $S_m$ is the compartment-A score of the TAD on the surface. $\overline{S_A}$ is the mean of all compartment-A scores. n denotes all compartment-B TADs that are located on the surface of the polymer conformation. $g_{BS}$ is the energy parameter for B-surface interactions. $S_n$ is the compartment-B score of the TAD on the surface. $\overline{S_B}$ is the mean of all compartment-B scores. To determine if a TAD is on the surface of the polymer conformation, we built a 3D convex hull from all monomer positions, and determined whether the TAD is one of the monomers on the surface of the hull.

The parameter used in the three sets of simulations were: For "B–B only", $g_{AA} = 0$, $g_{BB} = 50$, $g_{AS} = 0$, $g_{BS} = 0$. For "A–A and B–B", $g_{AA} = 50$, $g_{BB} = 50$, $g_{AS} = 0$, $g_{BS} = 0$. For "A–A, B–B, A–S, B–S", $g_{AA} = 50$, $g_{BB} = 50$, $g_{AS} = 40$, $g_{BS} = 25$.

**Bulk RNA-seq of E14.5 fetal liver**. Total RNA was extracted and purified from E14.5 fetal liver using the Qiagen RNeasy Mini Kit (Qiagen, 74104) according to manufacturer's instructions. Polyadenylated messenger RNA (mRNA) was selectively enriched using Invitrogen Dynabead mRNA Purification Kit (Invitrogen, 61006). A sequencing library was constructed with NEBNext Ultr II RNA Library Prep Kit (NEB, E7770S) and amplified with NEBNext Multiplex Oligos (NEB, E7335S). Paired-end reads were obtained with Illumina HiSeq2500. Paired-end reads were aligned to the mouse genome (Gencode vM14) with Tophat and transcripts were assembled with Cufflinks[64].

**Reporting summary**. Further information on research design is available in the Nature Research Reporting Summary linked to this article.

## Data availability

The genomic positions of TADs profiled in large-scale chromatin tracing were downloaded from http://chromosome.sdsc.edu/mouse/hi-c/download.html. Mouse E14.5 fetal liver Hi–C data were downloaded from GSM1718024. All chromatin traces, RNA profiles, lamina/nucleolar association data, and MATLAB programs for data analysis and simulation are available at https://campuspress.yale.edu/wanglab/MINA/. Raw imaging data are available from the corresponding author on request. Bulk RNA sequencing data

from this work are available at GSE148072. The source data underlying Figs. 1–6 and Supplementary Figs. 1–7 are provided as a Source data file. Source data are provided with this paper.

## Code availability

Data were collected using open source python codes from https://github.com/ZhuangLab/storm-control. All MATLAB programs for data analysis and simulation are available at https://github.com/SiyuanWangLab/MINA. Source data are provided with this paper.

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

## Acknowledgements

We thank Dr. Fred Sanford Gorelick, Dr. Yasuko Iwakiri, and Dr. Valentina Greco for helpful discussions and tissue samples for initial tests. This work is in part supported by Yale Liver Center Pilot Project Grant (awarded by National Institute of Diabetes and Digestive and Kidney Diseases of the National Institutes of Health under Award Number P30DK034989—Silvio O. Conte Digestive Diseases Research Core Centers) and the Yale Cooperative Center of Excellence in Hematology (YCCEH) Pilot Project Grant (awarded by the NIDDK under U54DK106857). S.W. is supported by NIH Director's New Innovator Award 1DP2GM137414. S.G.K. is supported by NHLBI Grant 1R01HL131793. M.L., B.Y., and M.H. are supported by the China Scholarship Council (CSC) Grants. J.S.D. R. is supported by NIH Predoctoral Training Grant (2T32GM007499). The content is solely the responsibility of the authors and does not necessarily represent the official views of the National Institutes of Health.

## Author contributions

S.W. conceived the study; M.L. and Y.L. performed experiments with help from B.Y., Y.C., J.S.D.R., M.H., and S.G.K. S.G.K. provided tissue samples. M.L., Y.L., and S.W. analyzed data. M.L., Y.L. and S.W. wrote the paper with inputs from B.Y., Y.C., J.S.D.R., M.H., and S.G.K.

## Competing interests

S.W. is one of the inventors on a patent applied for by Harvard University related to MERFISH. The remaining authors declare no competing interests.
