## [Peer Review File · Nature Communications]

Reviewers' comments:

Reviewer #1 (Remarks to the Author):

Liu et al combine DNA FISH (labeling mainly TADs on Chromosome 19), multiplexed RNA FISH (MERFISH) and immunofluorescence to correlate cell specific genome organizational patterns with nuclear landmarks like lamina and nucleoli in embryonic mouse tissue. Technically, this is an impressive manuscript although not completely novel. MERFISH and sequential DNA FISH has been combined in embryos before (see ORCA by the Boettigier lab). The main addition here is immunofluorescence to also correlate genome organizational patterns with association to nucleoli and lamina. The findings show that A/B compartments are cell-type specific and there is a very weak correlation between A/B compartment score and gene expression, there is some level of cell-type specific correlation between compartment score and nucleolar or lamina association and TADs are associated to chromosome surface instead of interior. These findings, while interesting, are not earth shattering. But overall this manuscript does demonstrate the type of rich data that can be obtained by combining many different imaging modalities and shows an interesting approach for genome organization studies moving forward. Therefore, I recommend its publication in Nature Communications. However, my major comment is regarding the modeling. The authors pitch the model as testing phase separation. I have a big problem with calling this phase separation. They are just introducing an interaction term between domains found in the same compartment. Why is this specific to phase separation? The nature of that interaction could be anything. I would support publication once authors take care of this point, which they can do mainly through rewording the text.

Reviewer #2 (Remarks to the Author):

Liu et al apply a multiplexed imaging method to obtain quantitative information on the organisation of chromatin in mammalian tissues.

To my opinion, the technique used in this work is very interesting and appealing and its use on tissues is notable and for sure deserves publication in Nature Comm.

Before publication, I have a few comments which I would like the authors to address:

1. The power of MINA clearly relies in identifying cell-specific chromatin arrangements within a real tissue yet it is unclear what are the main differences in terms of chromatin arrangements found in different cell types. Can the authors state these differences (if any) more clearly?
2. Can the authors quantify the geometric property of the chromatin tracings of TADs that switch compartment between different cells? Do you find TADs in B compartments generally smaller (in size or volume) than ones in A compartments?
3. The "polarisation" of compartments is interesting. Can the authors quantify if there is a specific arrangement of A with respect to B (or viceversa). For instance, can the authors compute a probability distribution function $g(x,y,z)$ representing the probability to find chromatin belonging to an A compartment at position $r=(x,y,z)$ with respect to the centre of a B compartment? This may be shown as a 3D density map (or 2D density map, if the precision along z is not good enough) with the distribution of A with respect to B. I think it may be an interesting plot to provide in the paper.
4. The authors perform simple polymer simulations to show that intra-chromatin interactions are not enough to explain the polarisation and organisation of TADs belonging in different compartments. They correctly identify that chromatin-lamina interactions are also important in this respect. I think that in this discussion two recent works should be mentioned, Falk et al Nature 2019 (cited by the authors) and also Chiang et al Cell Reports 2019. To the best of my knowledge these are the first to include the effect of the lamina on the spatial organisation of the genome. In particular, Chiang et al perform simulations of senescent cells, as mentioned by the authors as interesting cells to investigate using MINA.
5. It would be useful if the authors added cartoons of some key concept. For instance, in Fig.3, is it possible to add a small cartoon describing what they mean with polarisation? Also I think Figs 4 and 5 may benefit from cartoons to visually explain what they mean with "chromosome surface ratios", etc. A reader interested in the technical definition can then find it in the text.

Reviewer #1:

Liu et al combine DNA FISH (labeling mainly TADs on Chromosome 19), multiplexed RNA FISH (MERFISH) and immunofluorescence to correlate cell specific genome organizational patterns with nuclear landmarks like lamina and nucleoli in embryonic mouse tissue. Technically, this is an impressive manuscript although not completely novel. MERFISH and sequential DNA FISH has been combined in embryos before (see ORCA by the Boettgier lab). The main addition here is immunofluorescence to also correlate genome organizational patterns with association to nucleoli and lamina. The findings show that A/B compartments are cell-type specific and there is a very weak correlation between A/B compartment score and gene expression, there is some level of cell-type specific correlation between compartment score and nucleolar or lamina association and TADs are associated to chromosome surface instead of interior. These findings, while interesting, are not earth shattering. But overall this manuscript does demonstrate the type of rich data that can be obtained by combining many different imaging modalities and shows an interesting approach for genome organization studies moving forward. Therefore, I recommend its publication in Nature Communications. However, my major comment is regarding the modeling. The authors pitch the model as testing phase separation. I have a big problem with calling this phase separation. They are just introducing an interaction term between domains found in the same compartment. Why is this specific to phase separation? The nature of that interaction could be anything. I would support publication once authors take care of this point, which they can do mainly through rewording the text.

Response: We thank the reviewer for the very insightful and constructive comments, and for recommending publication of our manuscript in *Nature Communications*! We completely agree with the reviewer that we should replace “phase separation” with a more inclusive term in our manuscript. As the reviewer kindly pointed out, in essence, our model introduces self-associating interactions (A-A and B-B interactions) between domains of the same compartment to drive the separation of A/B compartments in space. To better convey this idea, in the revised manuscript, we have changed the terms “phase-separating interactions” to “self-associating interactions”, “phase-separating factors” to “self-associating factors”, and “phase separation” to “self-association”. Again we thank the reviewer for this very helpful suggestion!

Reviewer #2:

Liu et al apply a multiplexed imaging method to obtain quantitative information on the organisation of chromatin in mammalian tissues.

To my opinion, the technique used in this work is very interesting and appealing and its use on tissues is notable and for sure deserves publication in Nature Comm.

Response: We thank the reviewer for the enthusiasm about our work and for the support of our manuscript’s publication in *Nature Communications*! Below please find our point-by-point responses to the reviewer’s comments.

Before publication, I have a few comments which I would like the authors to address:

1. The power of MINA clearly relies in identifying cell-specific chromatin arrangements within a real tissue yet it is unclear what are the main differences in terms of chromatin arrangements found in different cell types. Can the authors state these differences (if any) more clearly?

Response: We apologize for the lack of a clear summary of the main differences of chromatin arrangements in the different cell types observed in our work. In the revised manuscript, we have added the following sentences to the Discussion as a summary of the findings on this front: “Using MINA, we observed three facets of chromatin arrangement that vary among cell types: 1) In terms of fine-scale chromatin folding, we identified a promoter-enhancer interaction in the *cis*-regulatory region of *Scd2* that

was enriched in fetal liver hepatocytes (Figure 2D-2F). 2) At a larger scale, we found that the folding schemes of TADs to A/B compartments also differ among fetal liver cell types, with some of the same TADs on the genomic map spatially assigned to different compartments depending on the cell type (Figure 3B and Supplementary Fig. 2). 3) In terms of the positioning of chromatin relative to other nuclear structures, we measured the spatial proximity of chromatin regions to nuclear lamina and nucleoli, and found that the lamina/nucleolar association ratios can systematically vary between cell types, e.g. between the closely related erythroblasts and proerythroblasts, independent of the compartmentalization differences (Figure 4A-4B).”

2. Can the authors quantify the geometric property of the chromatin tracings of TADs that switch compartment between different cells? Do you find TADs in B compartments generally smaller (in size or volume) than ones in A compartments?

Response: To experimentally address the reviewer’s question, we designed and purchased new oligonucleotide probe libraries to trace the sub-TAD folding conformations of three representative TADs that switched compartments between cell types. Specifically, we sub-divided TAD35, TAD46, and TAD49 into tens of consecutive 30-kb regions (36 regions for TAD35, 40 regions for TAD46, and 40 regions for TAD49), traced the sub-TAD folding conformations by probing these regions with MINA, and quantified the TADs’ levels of compaction through measuring the radii of gyration of the sub-TAD folding conformations. In Letter Figure 1, we plotted the TAD’s mean radius of gyration in each cell type versus the corresponding compartment score of the TAD in that cell type. The figure showed that none of the three TADs showed significant correlation between the level of compaction and the compartment score in this biological context.

It is possible that some other TADs will show a correlation between compaction and compartmentalization. It is also possible that this correlation or the lack of it depends on the biological context. We will continue to investigate this topic, but due to the significant cost of purchasing new probe libraries and the suspension of our research due to the current viral crisis, we would like to ask for the reviewer’s permission for us to continue this investigation in future studies.

Letter Figure 1. Mean radii of gyration of TAD49 (A), TAD46 (B) and TAD35 (C) versus their compartment scores in different cell types. Error bars indicate standard error. N: copy number. R: correlation coefficient. p: p value of the correlation. EC: endothelial cell. Hep: hepatocyte. Mac: macrophage. Meg: megakaryocyte. Ery: erythroblast. Pro: proerythroblast. Other: other cell types. Red and blue colors represent positive and negative compartment scores, respectively.

3. The "polarisation" of compartments is interesting. Can the authors quantify if there is a specific arrangement of A with respect to B (or vice versa). For instance, can the authors compute a probability distribution function $g(x,y,z)$ representing the probability to find chromatin belonging to an A compartment at position $r=(x,y,z)$ with respect to the centre of a B compartment? This may be shown as a

3D density map (or 2D density map, if the precision along z is not good enough) with the distribution of A with respect to B. I think it may be an interesting plot to provide in the paper.

Response: We thank the reviewer for this great suggestion. As the reviewer suggested, to visualize the probability distribution of compartment A regions with respect to compartment B, we calculated for each chromosome the centroid position of all compartment-A TADs (A centroid) and the centroid position of all compartment-B TADs (B centroid), and defined a vector from the A centroid to the B centroid as the polarization axis. Next, we rigidly rotated the chromosome so that the polarization axis is in the z direction, and we aligned the B centroids of all chromosomes at the origin. We then reconstructed a density map of compartment-A TADs with respect to the center of compartment B. Similarly, we generated a density map of compartment-B TADs with respect to the center of compartment A by aligning all A centroids at the origin. These figures are consistent with the polarization index metric in demonstrating the polarized organization of A-B compartments, and are now included as Supplementary Fig. 4 in the revised manuscript. And the following text has been added to the manuscript: “Consistently, when individual chromosomes were aligned along a vector that points from the center of compartment A to the center of compartment B, the distribution of compartment A TADs showed a significant displacement relative to the B center, and vice versa (Supplementary Fig. 4).”

4. The authors perform simple polymer simulations to show that intra-chromatin interactions are not enough to explain the polarisation and organisation of TADs belonging in different compartments. They correctly identify that chromatin-lamina interactions are also important in this respect. I think that in this discussion two recent works should be mentioned, Falk et al Nature 2019 (cited by the authors) and also Chiang et al Cell Reports 2019. To the best of my knowledge these are the first to include the effect of the lamina on the spatial organisation of the genome. In particular, Chiang et al perform simulations of senescent cells, as mentioned by the authors as interesting cells to investigate using MINA.

Response: We thank the reviewer for pointing out these very important papers that are highly relevant to our work. In the revised manuscript, we now discuss and cite both papers: “Particularly, recent studies have shown that the interactions between heterochromatin and nuclear lamina can drastically change global chromatin organization^{39,40}.”. Particularly for the Chiang et al paper, we state: “Tuning the balance among the four types of interactions could lead to alternative chromosome architectures, such as radially organized A/B compartments (Figures 6A and 6B), similar to the chromatin organization observed in senescence-associated heterochromatic foci⁴⁴ and consistent with the results from a recent simulation of senescent cells⁴⁰.”

5. It would be useful if the authors added cartoons of some key concept. For instance, in Fig.3, is it possible to add a small cartoon describing what they mean with polarisation? Also I think Figs 4 and 5 may benefit from cartoons to visually explain what they mean with "chromosome surface ratios", etc. A reader interested in the technical definition can then find it in the text.

Response: We thank the reviewer for the suggestions! In the revised Figure 3 and Figure 5, we have now added cartoons to help explain the concepts of polarization and chromosome surface ratio as the reviewer suggested.

REVIEWERS' COMMENTS:

Reviewer #2 (Remarks to the Author):

The authors have addressed all my comments. I am confident that this paper will be well received by the community. I congratulate the authors for the great work!

One thing to mention: the fact that they don't see a change in TAD gyration radius for different compartment scores is counter-intuitive if interpreted in the context of active/inactive chromatin. To my opinion, this is one of those "negative results" which are definitely interesting to publish and disseminate, and to pursue further in the future (when labs re-open). I suggest the author (if they wish) to give this result some space in the main text or SI.

Davide Michieletto

Response to reviewers

Comments from Reviewer #2:

The authors have addressed all my comments. I am confident that this paper will be well received by the community. I congratulate the authors for the great work!

One thing to mention: the fact that they don't see a change in TAD gyration radius for different compartment scores is counter-intuitive if interpreted in the context of active/inactive chromatin. To my opinion, this is one of those "negative results" which are definitely interesting to publish and disseminate, and to pursue further in the future (when labs re-open). I suggest the author (if they wish) to give this result some space in the main text or SI. -- Davide Michieletto

Reply:

We thank Dr. Michieletto for the inspiring comments and for the great support of our work. We agree with the reviewer that it is interesting that there is no significant correlation between the TAD gyration radius and its compartment identity. However, we hesitate to include it in the paper due to that these experiments have been done only once. Although we are confident about that these results should be reproducible, as all our previous results reported in the paper were reproducible in each replicate, we hesitate to report these new results in the paper because we have not formally performed a replicate due to the Covid-19 shut-down. We thank Dr. Michieletto for offering us both options regarding whether or not to include the new results. We have decided not to include them in this paper, but will continue the investigation in our next work.